# The Fabrication and Application Mechanism of Microfluidic Systems for High Throughput Biomedical Screening: A Review

**DOI:** 10.3390/mi11030297

**Published:** 2020-03-11

**Authors:** Kena Song, Guoqiang Li, Xiangyang Zu, Zhe Du, Liyu Liu, Zhigang Hu

**Affiliations:** 1College of Medical Technology and Engineering, Henan University of Science and Technology, He’nan 471023, China; kenasong@haust.edu.cn (K.S.); zu.xiangyang@163.com (X.Z.); duzhe@haust.edu.cn (Z.D.); 2College of Physics, Chongqing University, Chongqing 401331, China; guoqiangli1989@126.com (G.L.); lyliu@cqu.edu.cn (L.L.)

**Keywords:** microfluidic, scaffold, structure, material, mechanism, extracellular matrix (ECM)

## Abstract

Microfluidic systems have been widely explored based on microfluidic technology, and it has been widely used for biomedical screening. The key parts are the fabrication of the base scaffold, the construction of the matrix environment in the 3D system, and the application mechanism. In recent years, a variety of new materials have emerged, meanwhile, some new technologies have been developed. In this review, we highlight the properties of high throughput and the biomedical application of the microfluidic chip and focus on the recent progress of the fabrication and application mechanism. The emergence of various biocompatible materials has provided more available raw materials for microfluidic chips. The material is not confined to polydimethylsiloxane (PDMS) and the extracellular microenvironment is not limited by a natural matrix. The mechanism is also developed in diverse ways, including its special physical structure and external field effects, such as dielectrophoresis, magnetophoresis, and acoustophoresis. Furthermore, the cell/organ-based microfluidic system provides a new platform for drug screening due to imitating the anatomic and physiologic properties in vivo. Although microfluidic technology is currently mostly in the laboratory stage, it has great potential for commercial applications in the future.

## 1. Introduction

Microfluidic chips are extensively applied in in-vitro biomedical models because of their flexibility, such as having control over fluid and gas flow, and a chip architecture able to replace the classical 2D and 3D cell cultures [1]. Many organ models were established in recent years, based on the microfluidic platform, such as brain-on-a-chip, liver-on-a-chip, lung-on-a-chip, and breast-on-a-chip [2,3,4,5,6,7,8,9,10,11,12,13,14,15,16,17,18,19]. Body-on-a-chip is designed to include more than one organ on one chip [20]. Microfluidic chips provide a good platform for cell sorting and drug screening because of its high throughput and low cost. In cancer research, metastasis results in a 90% lethality of cancer, according to the available statistics [21], and hence, many in vitro models are established to mimic the natural status, in order to research the metastasis mechanism. So, the microfluidic chip technology is looked forward to as a high-efficiency strategy applied in biomaterials, cell responses, drug screening, and clinical treatment.

Design and production technology is the key to the function of the microfluidic system. The benefits of high throughput and the 3D microenvironment are additive in microfluidics, assisting its advantage in research and application. Diverse structures are designed, such as arrays and multi-channels. The mechanisms are also various to be used and based on acoustic, dielectrophoresis, and flow cytometry, to name a few. The mimicking extracellular matrix (ECM) in vitro is made from multifarious material, in which hydrogel, collagen, and Matrigel are the general choices [22]. In recent years, fluidic technology and its application are developing rapidly. However, it is rare for a review to focus on the fabrication, including the material and matrix microenvironment construction, and the application mechanism of the high throughput biomedical system. Thus, this review aims to summarize a high-throughput screening microfluidic chip in order to describe relevant examples, to benefit the implementation of: (1) the fabrication of high-throughput biomedical microfluidic systems (HTBMS), including the base material and method for fabricating the scaffold of the microfluidic chip with high throughput, and the construction of the matrix microenvironment in 3D HTBMS; (2) the application and mechanism of HTBMS, which is classified into two types by whether the matrix is contained in the system or not, such as biomedical synthesis, sorting, drug screening, and so on; (3) the future perspectives of microfluidic devices.

## 2. The Fabrication of High-Throughput Biomedical Microfluidic Systems (HTBMS)

### 2.1. The Scaffold Materials and Manufacturing Methods for HTBMS

The chemistry stability and thermostability of the scaffold materials are significant for HTBMS. For example, polymerase chain reaction (PCR) measurement needs a high temperature of 95 °C for DNA unchaining. For the high throughput application, the structures of multi-channels and the array of the special pattern are always fabricated in HTBMS. The manufacturing method and application are diverse, according to the properties of materials.

Glass and silicon are the initial materials of HTBMS because of their biocompatibility and convenient integration with electronics. Photolithography is used for the fabrication of patterns or channels. For example, Han et al. developed a glass-based microfluidic system by photolithographically etching, to detect DNA samples [23]. However, the high cost and the micromachining complexity limited its application. Silicon is also restricted for biomedical applications by its non-optical transparency, because the main biomedical analysis and detection methods are based on optical microscope technology [24].

Polydimethylsiloxane (PDMS) is a highly biocompatible material suitable for in vivo and in vitro. Plus, because of its high transparency, permeability, and stability, PDMS is a general material used to fabricate HTBMS for diverse applications, especially biomedical models, organ models, and drug screening [25,26,27,28]. PDMS is a liquid before being cured by high temperature, so it is always cured to form patterns by soft lithography. PDMS is always etched into either multi-channels or an array of a special pattern to build high-throughput models in HTBMS. Further, due to its softness, PDMS is a good choice for fabricating membranes to build organ models in HTBMS, such as lungs or hearts. For example, Agarwal et al., by combining the elastomer properties of poly(N-isopropylacrylamide) (PIPAAm) and the biocompatibility of PDMS, achieved the laminar architecture of the heart ventricle, which could mimic the diastolic and systolic stresses. The system is utilized for high-throughput pharmacological studies [29]. PDMS is widely used in the high-throughput biomedical microfluidic system. Xu et al., using PDMS, fabricated the arc-edge-channel monolithic valves for the large-scale integration of liquid manipulation and single-cell isolation with high throughput [30]. Liu et al. developed an electrospun fiber-embedded PDMS chip with high throughput to culture hepatocytes, in which hepatocyte spheroids were formed with different flow rates, providing a strategy to study drug metabolism in in-vitro liver tissue [31].

However, PDMS has some drawbacks, such as non-specific molecule adsorption, absorption of less hydrophobic molecules, incompatibility with many solvents and reagents, the release of uncrosslinked small PDMS molecules, and high cost. In order to avoid these problems, some other low-cost polymers were developed for HTBMS, such as polymethylmethacrylate (PMMA), cyclo-olefin copolymers (COC), thermoset polyester (TPE), polycarbonate (PC), polystyrene (PS), perfluoropolyether (PFPE), and so on, which have great development potential in microfluidic industry. These low-cost polymers allow for easy surface treatment and most of them are stable, transparent, and biocompatible, which is suitable for producing HTBMS [24]. Various fabrication techniques are available to create microfluidic networks in polymers, such as micromilling, hot-embossing, and injection molding. Some materials that are suitable for 3D printing have emerged, such as acrylonitrile-butadienestyrene (ABS), polyphenylenesulphone, polyamide (PA), Polyvinyl chloride (PVC), polyester, resin, and so on [32]. The following paragraph shows some examples of their typical application in HTBMS.

COC is commonly used in X-ray crystallography and is combined with the high controllability of microfluidic environments. The fields of biophysics and biology have benefited enormously from such approaches. A general approach of screening for crystal growth and data collection is to use a combination of COC and PDMS to allow the inclusion of a microfluidic valve and use silane to bond the layers [33,34,35,36,37]. Some other researchers use COC to fabricate an all-COC device. Denz et al., using soft lithography, produced a channel structure that was established using an all-COC microfluidic device to measure X-ray inflow with high throughput. They investigated the early time points of the assembly of vimentin intermediate filament proteins into higher-order structures using the aforementioned system [37]. A perfluoropolyether (PFPE) biochip was built to culture and analyze the cells of liver and kidney (shown in Figure 1B). Two patterned layers of precise and regular microchannels in the microdevice were developed by the photocuring of perfluoropolyethers (PFPEs) and the layers were sealed by UV irradiation [38]. Polycarbonate (PC), which is available for performing an allele-specific reaction, has a high glass transition temperature. Soper et al. combined a PMMA microarray with PC substrates to fabricate, using UV modification protocols, with integration into microfluidic platforms for the detection of low-abundant DNA point mutations [39]. Shisheng et al. developed a novel one-step method that combines bonding and hydrophobic surface modification processes to rapidly manufacture droplet-generating chips using an all-PC device. The patterns on PC substrates were fabricated using an injection molding machine with a high-precision nickel mold (shown in Figure 1A) [40]. Yeung et al. developed a microfluidic device with a hollow microneedle architecture using resin by 3D-printing for the purpose of transdermal drug delivery [41]. Further, TPE is an advanced material for microfluidic fabrication because it is highly reproducible and is capable of sealing to other materials, and is especially able to bond with glass [42]. TPE could be successfully sealed to dielectric, metallic mirrors, in which PDMS does not form a permanent bond. Polyurethane Methacrylate (PUMA) and Norland Adhesive 81 are also the materials suitable for rapid prototyping when bond strength, or transitioning to commercialization are important goals [43].

### 2.2. The Construction of 3D Microenvironment in HTBMS

For the cell-based 3D HTBMS, the extracellular matrix (ECM) is an important component of the system, because the good cell survival and growth status is the key precondition for the functional measurement, especially for high-throughput cell/organ-based drug screening. The 3D microfluidic system, including ECM, which is the junction structure neighboring cells in vivo, provides a platform to mimic the real 3D microenvironment for biomedical models. The experiments in 3D HTBMS have been implemented on a diverse range of systems and applications, such as microfluidics-based 3D cell culture systems, organ-on-a-chip systems, and drug screening platforms [44,45,46]. In general, ECM in HTBMS performs three functions in the system: (1) providing a real-like microenvironment in HTBMS for cell growth, even differentiating organ structures; (2) building drug gradients in HTBMS, utilizing the high penetrability of ECM; (3) the ECM is cured into an array with a special pattern for building high-throughput cell aggregation models, spheroid models, and even tissue/organ models.

#### 2.2.1. The Microenvironment Construction of 3D HTBMS for Cell Culture

The fate of cells in vivo is affected largely by the external microenvironment, including physical and chemical factors, cell–cell interaction, and cell–ECM interaction [47]. So, mimicking the real ECM is significant in the 3D HTBMS for cell cultures. The perfected materials are collagen, Matrigel, elastin, glycoproteins, and polysaccharides, due to most of them being components of the in vivo ECM [45,48,49]. In fact, polymer hydrogels have become the raw materials of the 3D cell microenvironment due to their similar physicochemical properties (nutrients, pH, oxygen, hardness, and morphology) and good biocompatibility. According to the difference model, the components and properties of ECM are controlled in the in vitro system.

Park et al., using collagen as the ECM, developed an injection molded plastic array device to culture lymphocytes and cancer cells for assessing the killing abilities of cytotoxic lymphocytes in a 3D microenvironment through spatiotemporal analysis. They found that 3D ECM showed lower cytotoxicity compared with the 2D microenvironment significantly [50]. Sung et al. constructed a reliable 3D collagen culture platform in microfabricated systems to investigate reciprocal interaction between the extracellular matrix (ECM) and cells under various conditions. Array-based microchannels were employed to provide more replicates with less sample volume than conventional means [51]. Lii et al., using Matrigel as the ECM, developed a real-time, dynamic control microfluidic system for the study of 3D cell cultures. The dynamic control is based on the reagents perfusion of the 3D environment of cells [52]. Lanz et al. perfused ECM composition (Matrigel, BME2rgf, collagen I) as the ECM into the microfluidic channel to culture tumor cells for selecting therapies [53]. For mimicking the real ECM, tissue ECM, which is processed by removing soluble tissue components and fragmenting the matrix, is used to build HTBMS. Beachley et al. used tissue ECM to incorporate with cells to generate 3D cell–matrix microtissue arrays to investigate the responses of human stem, cancer, and immune cells to tissue ECM arrays originating from 11 different tissues with high throughput [54].

The artificial polymeric hydrogels as an artificial cellular microenvironment have a pivotal advantage due to their flexibly-controlled properties. The properties are easily tuned by the crosslinking density in the 3D network of the microstructure of the polymeric hydrogels, such as stiffness, swelling ratio, and porosity. So, various hydrogels have been developed for fabricating the 3D engineered matrix. Skardal et al. utilized a UV-crosslinkable hydrogel solution as the ECM to culture liver cells in a microfluidic device. Using the platform, they assessed albumin, urea production, and αGST release successfully [55]. Karamikamkar et al. mixed alginate with collagen as the ECM of breast cancer cells to prepare homogeneous cell–hydrogel beads. The alginate acted as a fast gelling component during bead encapsulation to prevent the collagen gel from forming chunks caused by sensitivity to temperature. The microfluidic system could develop tumor spheroid models with high uniformity and throughput [56]. Using alginate hydrogel as the ECM component, Alessandri et al. developed a 3D-printed co-extrusion device, allowing cell encapsulation within hydrogel hollow spheres and 3D culture (shown in Figure 2B) [57].

#### 2.2.2. The Microenvironment Construction of 3D HTBMS for Organ-on-a-Chip

For organ-on-a-chip HTBMS, its special feature is forming the organ-like structure and function. The matrix microenvironment of cells is important to build a real-like organ model for cell adhesion, support cell growth, and to enhance cell–matrix interactions. However, some organ-on-a-chip technologies are labor-intensive and lack an extracellular matrix (ECM). The organ-on-a-chip with a matrix microenvironment attracted more attention and has developed in recent years [58]. In addition to the traditional methods of perfusion, drip coating, and mixing with cells, the new methods of droplets and bio-ink-based bioprinting have also become important technologies used to introduce ECM into organ models. However, to mimic different organs or different states of the same organ, the selection and ratio of the extracellular matrix, as well as the location, are different. Here, we only introduce liver, lung, mammary, and angiogenesis as examples.

Hegde et al. fabricated a two-chambered device including a porous polyethylene terephthalate (PET) membrane between two layers of PDMS. Collagen as the ECM mixed with hepatocytes were perfused into the bottom chamber, meanwhile the top chamber was accessible for flow. The device is used to mimic the structure and function of hepatic lobules [59]. Lee et al., using 3D cell-printing, developed a 3D liver-on-a-chip with liver decellularized ECM bio-ink for a 3D microenvironment and vascular/biliary fluidic channels [58].

Huh et al. produced a lung-on-a-chip using a 10 μm PDMS membrane with ECM separating the chip into two regions. The upper and lower regions are used to culture alveolar epithelial cells and human pulmonary microvascular endothelial cells, respectively, to mimic the alveolar–capillary barrier. The structure of the coated ECM membrane was altered by a vacuum to mimic the expansion and contraction of the alveoli during respiration [1]. Based on Matrigel promoting monolayer formation and being the main containing protein of the basement membrane, Humayun et al. employed Matrigel by mixing Type I collagen as the ECM of smooth muscle cells and epithelial cells in a lung-on-a-chip. The mixing of the hydrogel was suspended between the smooth muscle cell layer and epithelial cell layer to separate the top and bottom compartments to mimic the lung airway tissue microenvironment in a microfluidic device [60].

Fan et al. fabricated microchamber arrays in collagen to mimic the matrix microenvironment of the mammary gland to investigate the interactions between invasive breast cancer cells and stromal cells (shown in Figure 2A). The tunable biochemical gradients are achieved in the microfluidic system due to the permeability of collagen [61].

Du et al. developed a microfluidic droplet array-based cell–coculture system for building angiogenesis in droplets. In the system, Matrigel is used as the ECM based on the differentiation-promoting properties of Matrigel (shown in Figure 2C) [62].

## 3. The Application and Mechanism of HTBMS

Microfluidic devices have been widely used in biological applications, such as drug synthesis, medical diagnostics, DNA and protein analysis, and drug development. That is due to their many advantages, especially miniaturization, which allows for rapid analysis using portable instruments. From the structural characteristics of the system, it can be divided into two types: (1) the biomedical synthesis and separation in HTMS without a matrix environment; (2) the application of droplet and cell/organ-based drug screening of HTBMS with a matrix environment. The following describes the applications and the mechanism from these two aspects.

### 3.1. Biomedical Synthesis and Separation in HTMS without Matrix Environment

#### 3.1.1. Biomedical Synthesis and Detection

The controlled synthesis of nano/microparticles, such as core/shell, Janus, nanocrystals, liposomes, and biopolymeric microgels, is an important biomedical application of HTBMS. The injection and mixing mechanism lead to self-assemblies for rapid chemical reactions and a highly-ordered structure, becoming the new alternative for biomedical synthesis and detection, and that strongly depends on the channel structure [63,64].

Y-shaped and T-shaped channels, multi-flow focusing junctions, and three-dimensional flow focusing in aligned capillary tubes are the typical structures for injection. The shear force, local concentration of reagents, and injecting rate are controlled in the special channels. Through the control of flow conditions, the size is easily varied, and the droplet formation could be very fast at a high flow rate. Wang et al. integrated complex Y-shape and T-shape channels to combine automated sampling, dilution, metering, and mixing, resulting in a combinatorial library composed of 648 different DNA-encapsulated supramolecular nanoparticles within 2.5 h [65]. Ran et al. used a multi-flow focusing junction to make multifunctional liposomes in the microfluidic system. Liposomes were prepared by injecting the lipid-containing isopropanol solution to the central channel and squeezed by the two vertical streams of phosphate buffer solution (PBS) from the vertical channels. By tuning the flow rate ratio and the ratio of the lipids to lipids and to folate, the size and surface properties of these liposomes could be controlled precisely [66]. Pedro et al. developed a fully integrated microfluidic device with a multi-inlet to combinatorial synthesis and optimize the targeted nanoparticles for cancer therapy [67]. 

Rapid mixing in a microfluidic chip is the other aspect of biomedical synthesis and detection. Mixing is normally dominated by diffusion instead of turbulence because the flow on a microscale has a low Reynolds number. However, mixing driven by diffusion is relatively time-consuming and inefficient. So it is critical to design the structure of the microfluidic system, especially involving reaction, drug gradient, and so on [68]. The traditional mixing method is to increase the length of the channel to increase the mixing path [69]. Fernández-Carballo et al. established a microfluidic, real-time, fluorescence-based, continuous-flow reverse transcription PCR system by fabricating a channel 95 mm long at the mixing section for facilitating the mixing of the fluids from both inlet ports [70]. To enhance mixing efficiency, a variety of schemes have been designed. The most typical structures are the channel of herringbone and Tesla structure. A Tesla structured channel is a delicate design that provides multiple diffusion domain regions and convection domain regions for both fully mixing and the specialty of a single-direction guide flow. So, a Tesla structure is a typical structure in a passive micromixer [71]. Chien-Chong et al. explored Tesla structure (Figure 3B) with a simple in-plane structure in a passive microfluidic mixer, realizing an excellent mixing performance at a high flow rate. Further, its pressure drop was less than 10 KPa at the flow rate of 100 mL/min [72]. Yang et al. reported a high-performance micromixer with three-dimensional Tesla structures to realize the effective mixing process in the microfluidic devices for binding anti-rabbit IgG-CFL555 on a cancer biomarker of EGF receptor L858R mutant specific rabbit. The proposed micromixer has illustrated excellent mixing performance for Reynolds numbers ranging from 0.1 to 100 (0.015–15 μL/s) with its pressure drop less than 1054 Pa at the Reynolds numbers of 100 [68]. Duarte and Guevara, et al. designed a microfluidic device with Tesla structure integrating three labeling stages of mixing, incubation, and crosslinking within the microdevice, performing viable qPCR and qLAMP for *Salmonella typhimurium (S. typhi) and Escherichia coli (E. coli)* O157. The device could detect viable bacteria with a limit of detection of 7.6 × 10^3^ and 1.1 × 10^3^ CFU/mL for *S. typhi* and *E. coli* O157, respectively [73]. The herringbone structure is the other typical structure for mixing. It is based on patterns of grooves on the flow of the channel and can generate steady chaotic fluid in microchannels. The chaotic flow is produced by subjecting volumes of fluid to a repeated sequence of rotational and extensional local flows, while the sequence of local flows is achieved by varying the shape of the grooves in the herringbone mixer [74]. For example, Streck et al. reported a microfluidic device with a staggered herringbone mixer (Figure 3A) to formulate cell-penetrating peptides-tagged poly(lactic-co-glycolic) acid nanoparticles [75].

#### 3.1.2. Biomedical Sorting and Diagnostics

Biomedical sorting and diagnostics are the significant applications of HTBMS. The advance of high throughput in microfluidics provides the possibility of various novel diagnostic and therapeutic applications using microfluidics technologies. Many diagnostics methods are rapidly developed, such as genetic analysis and protein analysis, due to its beneficial and essential application to both precision medicine and personalized medicine. The mechanisms and application of cases of sorting and diagnostics are diverse. For example, Garcia-Cordero et al. developed a platform combining microarrays and microfluidic techniques to measure four protein biomarkers in 1024 serum samples for cancer diagnostics based on a surface fluorescence sandwich immunoassay [76]. Simultaneously, microfluidics, based on structure and external fields, has also developed rapidly for biomedical sorting and diagnostics, such as microfiltration, dielectrophoretic, magnetophoresis, acoustophoresis, biomimetic separation, and integrative techniques. The throughput and efficiency have also become the indicators of device evaluation. Table 1 collected many cases to display the throughput and efficiency to capture circulating tumor cells (CTCs) based on diverse mechanisms in detail.

The traditional method for particle separation is utilizing special geometry structure in microfluidics, such as filter elements and physical barriers, without introducing an external field. The main technique is controlling the structure of the porous materials, barriers, array, and tuning the shape, size, and quantity [77,78,79,80,81,82,83]. For example, circulating tumor cells could be enriched from peripheral blood by tuning the size of the microfilter because the deformability and size of the nucleus dominate cellular translocation through microconstrictions under a normal physiological pressure range [81]. Warkiani et al. fabricated a multi-layer polymeric microsieve which includes narrow slot pores, detecting waterborne pathogens at low concentrations [79]. Yunlang et al. fabricated a porous scaffold with a porous microstructure and conforming to the shape of the uterine cavity (shown in Figure 4A), to load and control the low release of a drug for the improvement of postoperative severe intrauterine adhesions [78]. VanDelinder et al. used the structure of the filter (weir or pillar type) to separate the microsampling of blood based on the sizes and weights of the cells in blood [82]. Yoon et al. presented a microfluidic device with a slanted weir, which could separate circulating tumor cells from the unprocessed whole blood utilizing the size and deformability peculiarity [83].

Dielectrophoresis is a label-free method to manipulate particles. The neutral particles are first polarized under the action of an AC or DC electric field, and then the polarized particles are subjected to a dielectrophoretic force and move in a direction along with the electric field strength changes. Embedding dielectrophoresis in microfluidic devices greatly expands the microfluidic application. This method is non-destructive for cellular membranes, which is a benefit for the downstream characterization [84,85,86,87,88]. For example, circulating tumor cells (CTCs) are separated due to the capacitances of the malignantly transformed cellular membranes that differ from other cells [85]. So the research of using dielectrophoretic on biological analysis has become a hot lab-on-chip research topic, such as cell manipulation and cell isolation (shown in Figure 4B). The separation target is not limited to cells, but can also be applied to purifying nano-scale virus, nanoparticles, DNA, and so on [88]. Park et al. utilized dielectrophoresis to manufacture single-cell traps in a microfluidic chip for the analysis of suspended cancer cells [89]. Sun et al. pre-created patterned dielectrophoretic force in a microfluidic device to direct cells into a capture zone. The device was highly efficient in capturing the rare circulating tumor cells from blood samples [90].

Magnetophoresis utilizes magnetic levitation under a non-uniform magnetic field that combines magnetic and hydrodynamic forces in a microfluidic device to manipulate particles/cells. Labeled free cells are placed in a uniformly magnetic media. The magnetic media is attracted by the gradient magnetic field, while the cells in the uniformly magnetic media will be pushed away preferentially. The procession causes cells to be manipulated continuously [91,92]. So the crucial factors are the magnitude and non-uniformity of the magnetic field, particle/cell magnetic susceptibilities, and suspension media. Because of its advantages of high penetrability, low-heat generation, being gentle, and non-destructive for proteins or peptides, magnetophoresis is greatly utilized to manipulate cells, such as cell separation, trapping, and measurements, when combined with microfluidic technology [93,94]. Zhao et al. made a biocompatible ferrofluid as the media to separate a variety of low-concentration (~100 cancer cells/mL) cancer cells with a high throughput of 1.2 mL/h from undiluted white blood cells. The illustration is shown in Figure 4D. The separation method is efficient and has a high purity for a variety of cancer cells, such as A549 lung cancer, H1299 lung cancer, MCF-7 breast cancer, MDA-MB-231 breast cancer, and PC-3 prostate cancer cell lines [92]. Shamloo et al. utilized negative magnetophoresis on a rotating disk to sort the cells into multiple outlets with a throughput of 3100 cells/s. This device reduced the exposure time of the cells inside the ferrofluid combining the responses of magnetic buoyancy force and centrifugal force, which improves the cell viability [95]. Sun et al. developed a microfluidic magnetic deposition system separating the intrinsically magnetic spores and red blood cells for microscopic analysis. Guided by this system, the trajectories of the cells have enabled the analysis with a throughput of 1.2 mL/h as well as more than 95% of intrinsically magnetic bacillus spores were separated [96]. However, the intrinsic properties of the cells are sometimes not enough to be separated by magnetophoresis. The cells could be labeled by special antibody-magnetic particle conjugates to increase the magnetic susceptibility [94,97,98,99,100,101]. Schneider et al. labeled cells with antibodies conjugated to magnetic nanospheres. The method gives rise to the proportional relationship between the number of the attached magnetic nanospheres on a cell and the cell surface marker number. The magnetophoretic mobility of the labeled cells in the gradient magnetic field enables the potential fractionation of the cell population [98]. Gourikutty et al. labeled the cells with customized mixture formed by antibody complexes with magnetic particles, exploiting the difference in properties for separation under a very-high gradient magnetic field [94].

In an acoustic wave, particles are pushed to either the nodes or the anti-nodes, depending on the pressure distribution of the acoustic wave. The magnitude of the acoustic force is determined by various factors, including the size and elasticity of particles, the surrounding media, and the amplitude and frequency of the acoustic waves. Because of its simplicity, low-cost fabrication, biocompatibility, label-free, and rapid, localized fluid actuation, acoustophoresis is used for cell manipulation and detection, such as cell isolation, cell sorting, and aggregation [102,103,104,105]. Jung et al. developed an acoustofluidic device, able to control the nodal position by exploiting differences in specific gravity and ultrasonic velocities. Based on the device, the particles could be separated with high throughput and high selectively by moving the nodal band closer. The T lymphocytes are separated at a high velocity of 600 μL/min by altering the net sonic velocity to reposition acoustic pressure nodes [106]. Urbansky et al. optimized the buffer conditions of an ultrasound-based microfluidic device for changing the acoustophoretic mobility of the cells to separate mononuclear cells from red blood cells, the two kinds cells of which behave similarly in an acoustic standing wave field. The separation technology enables gentle, continuous, and high-throughput separation and enrichment of mononuclear cells from the label-free whole blood [107]. Wu et al. integrated acoustics and hydrodynamics into three-dimensional acoustofluidic tweezers (shown in Figure 4C), which enables the separation of microparticles and cells into multiple high-purity fractions with a wide range. The system can separate the microparticles of 10, 12, and 15 micron size, as well as the cells of erythrocytes, leukocytes, and cancer cells with a high throughput of 500 μL/min [108].

Single-parameter separation limits the resolution within the similar signals between cells. The integration of multiple separation forces enhances the distinguishable parameters. Microfabrication techniques offer the possibility to integrate multiple separation forces in a microfluidic device in a non-interacting manner, which enhances the application of the microfluidic device. Adams et al. integrated acoustic and magnetic separation forces in a monolithic device (shown in Figure 4E) for high-purity and high-throughput multi-parameter particle separation. The device was used to separate a multi-component particle mixture with high purity at a throughput of up to 10^8^ particle/h. The device is predicted to be suitable for a wide range of cells, from bacteria to mammalian cells. However, the particles need to be labeled by magnetic nano/microparticles in the device because of the magnetophoresis separation [109]. Liu et al. combined microfluidic deterministic lateral displacement and affinity-based cell capture in a microfluidic platform with continuous and high throughput. Using the device, breast cancer cells were isolated from spiked blood samples with a throughput of 9.6 mL/min [110].

### 3.2. The Application of Droplet and Cell/Organ-Based Drug Screening of HTBMS with Matrix Environment

Although there are advantages and applications of HTBMS without matrix environment, especially in biomedical synthesis and sorting, however, this system suffers from some limitations, such as the restricted simulation of the anatomic and physiologic properties in vivo. The HTBMS with a matrix offers the opportunities for biomedical application at the cell level under physiologically relevant conditions, such as drug screening by cell response and organ metabolism. Introducing a matrix into the system also provides a way to fabricate cell encapsulation, which is useful for cytotoxicity testing and as the bio-ink for bioprinting.

#### 3.2.1. Hydrogel Droplet for Biomedical Cell Encapsulation and Analysis

Hydrogel droplet microfluidics adds the intercellular interaction and local environment control while retaining a high throughput compared with flow cytometry. Many kinds of hydrogels were widely exploited for the microfluidic system, including collagen, gelatin, alginate, polyethylene glycol (PEG), and poly(acrylic acid) (PAA) based hydrogels. As for the hydrogel processing techniques, a droplet microfluidic system is well suited, such as droplets formed by the driving force of a microfluidic system and cured by photopolymerization. Single/several cells could be individually encapsulated in monodisperse microdroplets, for further manipulation and analysis by controlling the local environment precisely. A matrix in a droplet could support the long-term cell culture by mimicking the real extracellular environment, which is beneficial for further heterogeneity analysis, such as cell proliferation, differentiation, and metastasis at the single-cell level [113,114,115,116,117,118].

Zhang et al. developed a device integrating encapsulation, manipulation, and detection for phospholipid profiling of cell membranes. Single cells are separated and encapsulated with a matrix to form droplets. This system could provide the physiological context for the live single cell with well-preserved lipid information. Using the system, the heterogeneities between the normal cells and cancer cells were caught, meanwhile, the heterogeneity of the same cells before and after the drug treatment changed obviously [119].

Linsenmeier et al. developed a droplet-based microfluidics by introducing a hydrogel matrix to mimic the cytoskeleton for studying the dynamics of synthetic membraneless organelles of artificial cells (shown in Figure 5A). They observed the nucleation, growth, and coarsening in volumes comparable to cells, and they found that the timescale of phase separation decreases linearly as the volume of compartment increases [120].

Lee et al. displayed a flow-focusing droplet microfluidic device to build a tumor spheroid 3D culture model supported by bioactive spherical microgels. The uniform-sized cell-laden microgels spheroids were fabricated by photocrosslink, and the mechanical properties could be controlled by the concentration of a gel-forming polymer. Using this system, they explored the effect of mechanical properties of microgels on cell proliferation and the eventual spheroid formation [121].

#### 3.2.2. Cell/Organ Based Drug Screening

Cell/organ-based HTBMS provides a good platform for drug screening because it could imitate the anatomic and physiologic properties, including the properties of the specific cells and the function of organs in an in vitro system, and decreases the cost and time compared with the traditional animal testing. The platforms allow for cell growth in 3D ECM such as hydrogels, droplets, and scaffolds to mimic the real microenvironment conditions in vivo for drug screening by cell response and organ metabolism. It integrates diverse technologies, automatic operation, and microsized scale, which supports many benefits, including being low cost, high-resolution live imaging, having a high throughput, and efficiency in drug screening [122,123,124,125,126,127]. Furthermore, it could be used to investigate the effects or cytotoxicity at multiple doses in parallel and combination drug screening. These benefits are enabled by the structural flexibility and high permeability of the ECM in HTBMS [128].

Chang et al. developed a multi-layer microfluidic device integrating 3D matrix with microchannels, porous membranes, and microvalves to produce a 3D cell culture-based combinatorial drug screening array. They performed a combination drug screening of two anti-cancer drugs (doxorubicin and paclitaxel) on the breast cancer cell lines of MDA-MB-231 and MCF-7 for demonstration (shown in Figure 5B). The device is confirmed by building a combinatorial gradient successfully for multiple drug screening with reducing the operation time and the number of samples [129].

Chen et al. developed a breast-on-a-chip including multicellular tumor spheroids, microvessel walls, and the extracellular matrix to rapidly screen drugs and drug delivery, and to predict the in vivo performance. The workflow is shown in Figure 5C. It reveals that a carbon dots-based drug delivery system could be transported across an endothelial monolayer within 3 h and was non-toxic to the cells throughout the experiment. The capabilities of the drug delivery system were performed efficiently in spheroids of triple-negative breast cancer representative BT549 (TNBC) and non-TNBC representative T47D spheroids. [16]

Koo et al. reported a microfluidic brain model including an extracellular matrix and a blood–brain barrier to measure the organophosphate-based compounds (OPs) effects on barrier integrity and acetylcholinesterase (AChE) inhibition. They found that the OPs penetrate the blood–brain barrier and inhibit AchE activity rapidly. The drug toxicity showed in this brain-on-a-chip model had reasonably good correlations with the available in vivo data [130].

Lee et al. developed a 3D liver-on-a-chip with a liver decellularized ECM as the 3D microenvironment and vascular/biliary fluidic channels. They demonstrated that the system is promising in vitro liver test platform for drug discovery based on the evaluated effective response of acetaminophen [58].

## 4. Conclusions and Outlook

Microfluidic technology has been rapidly developed in recent years. A myriad of microfluidic techniques has been explored, from microfluidic fabrication to the operating mechanism. It has a good application prospect because of its small sample volumes and high throughput properties, especially in drug mixture preparation, cell sorting for classifying aggregation, cell-based drug screening, and so on. In this review, we focus on the fabrication and application mechanism of HTBMS without ECM and HTBMS with ECM, including the base scaffold materials, the microenvironment construction of 3D HTBMS, and the application and mechanism of HTBMS without, and with, ECM. This review collects a large number of cases to classify and analyze the characteristics to provide a guide for choosing the appropriate fabrication and mechanism for different applications. However, up until now, many microfluidic systems are still in the laboratory stage and have not yet been widely used for commercial applications. Combined with the current development situation, two suggestions are provided: (1) Developing the microfluidic platform of multi-parameter complex mechanism, which could not only improve the accuracy of the instrument but also broaden its application range; (2) for drug screening, developing the microfluidic platform containing 3D microenvironment, which will greatly improve the reliability of test results and be suitable for primary cell personalized drug testing. We predict that microfluidic technology will certainly make commercial applications widespread in the biological and medical fields.

## Figures and Tables

**Figure 1 micromachines-11-00297-f001:**
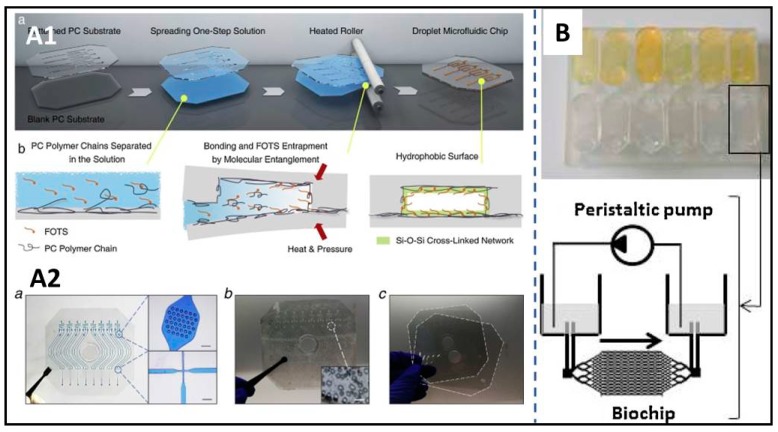
Two exhibitions of diverse materials are used to create a microfluidic system with high throughput. (**A1**,**A2**) A polycarbonate-based droplet microfluidic chip was fabricated by one-step bonding and hydrophobic surface modification, reproduced with permission from [40]. (**B**) Perfluoropolyether (PFPE) and polydimethylsiloxane (PDMS) connected under the integrated dynamic cell cultures in microsystems (IDCCM) bottom layer to fabricate a fluorinated microfluidic biochip to culture liver and kidney cells, reproduced with permission from [38].

**Figure 2 micromachines-11-00297-f002:**
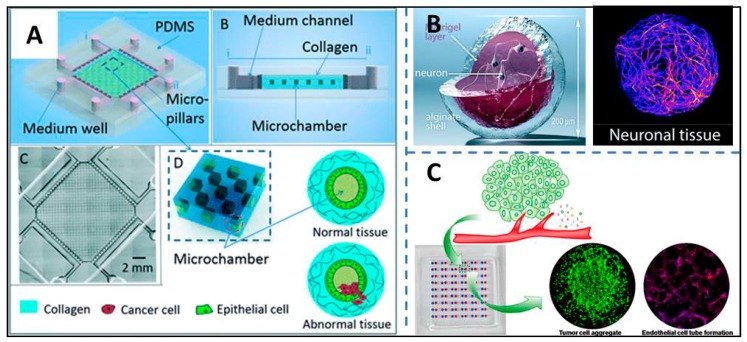
National and polymer hydrogels are utilized to mimic the extracellular matrix (ECM) in microfluidic systems. (**A**) A biomicrofluidic system, using collagen as the ECM, revealed the interactions of tumor and stroma, reproduced with permission from [61]. (**B**) A 3D-printed microfluidic device used alginate hydrogel to produce functionalized hydrogel microcapsules embedding human neuronal stem cells, reproduced with permission from [57]. (**C**) A high-throughput tumor angiogenesis assay was built on the ECM of Matrigel, reproduced with permission from [62].

**Figure 3 micromachines-11-00297-f003:**
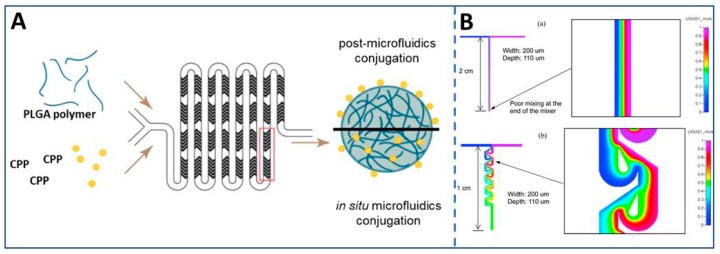
Two typical structures for mixing adequately in a microfluidic system. (**A**) A microfluidic system utilized herringbone structures to complete a zero-length crosslinking reaction for the covalent attachment of cell-penetrating peptides (CPPs) to poly(lactic-co-glycolic) acid (PLGA) nanoparticles, reproduced with permission from [75]. (**B**) Comparing the mixing efficiency of the T-type and the Tesla structure, the Tesla structure mixes the two fluids highly-efficiently through a 1 cm length of the perpendicular channel, relative to the T-type channel that shows a weak mixing efficiency, even though it flows through a 2 cm perpendicular channel, reproduced with permission from [72].

**Figure 4 micromachines-11-00297-f004:**
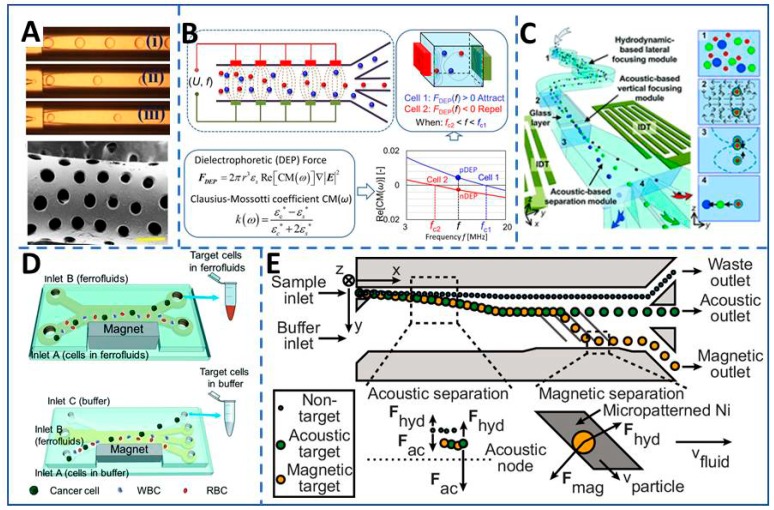
The application and mechanism of the microfluidic system without matrix for biomedical sorting with high throughput. (**A**) The filtering effect of porous scaffolds was used to dosing for the prevention of intrauterine adhesion, reproduced with permission from [78]. (**B**) Electrodes were embedded in a microfluidic device to manipulate cells by dielectrophoresis, reproduced with permission from [88]. (**C**) Acoustofluidic tweezers were used to focus and separate microparticles and cells into multiple high-purity fractions, reproduced with permission from [108]. (**D**) Magnetophoresis was used to separate cancer cells from blood cells in ferrofluids, reproduced with permission from [92]. (**E**) A microfluidic device integrated acoustics and magnetics to separate particles by multiple parameters, reproduced with permission from [109].

**Figure 5 micromachines-11-00297-f005:**
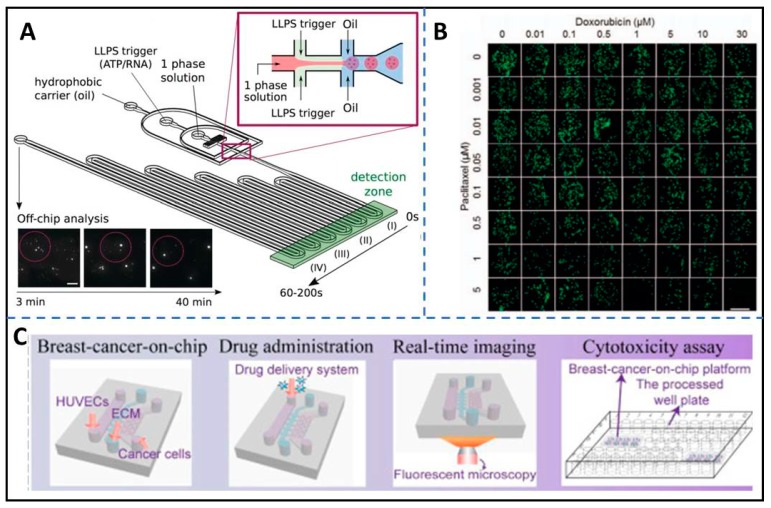
The biomedical application of high-throughput biomedical microfluidic systems (HTBMS) with ECM. (**A**) A droplet-based microfluidic system was developed by introducing a hydrogel matrix mimicking the cytoskeleton to synthetic membraneless organelles, reproduced with permission from [120]. (**B**) A combinatorial drug screening system was developed based on 3D-cultured cells. The figure shows the representative fluorescent micrographs of cells after 24 h combinatorial treatment with two anti-cancer drugs (doxorubicin and paclitaxel), reproduced with permission from [129]. (**C**) A breast-on-a-chip is fabricated using basement membrane extract (BME) as the ECM for therapeutic evaluation of drug delivery with high throughput. The workflow is displayed in the figure, reproduced with permission from [16].

**Table 1 micromachines-11-00297-t001:** The display and compare the throughput and efficiency of diverse mechanisms for capturing circulating tumor cells (CTCs) from mimicking/real blood samples.

Mechanism	Application	Throughput	Efficiency
Filtration	Capture LM2 MDAMB- 231 breast cancer cells from phosphate-buffered saline [83]	2.5 mL/h [83]	97.1% [83]
Dielectrophoretic	Capture CTCs from DEP buffer and patient blood [90]	1 mL/h and 8 mL/h respectively [90]	~100% and 28.3% ± 7.6% respectively [90]
Magnetophoresis	Capture cancer cell MCF-7 from PBS, and whole blood [111]	1 μL/min [111]	95.8% and >94% respectively [111]
Acoustophoresis	Separate tumor cells from white blood cells in Blood [112]	70 μL/min [112]	97.4%–98.4% [112]
Integrative techniques	Capture MCF-7cells from 10× diluted blood samples [110]	9.6 mL/min [110];	90% [110]

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
