# Peer review of "The Fabrication and Application Mechanism of Microfluidic Systems for High Throughput Biomedical Screening: A Review"

_micromachines, 2020, doi:10.3390/mi11030297_

Round 1
Reviewer 1 Report
In this review paper, the authors summarized research effort in the field of high throughput microfluidic chips from three aspects: (1) the material of the fluidic chips (2) mixing and sorting mechanisms (3) the materials of 3D ECM commonly used in microfluidic chips. Some recent developments are included, and new applications are discussed. This review provides certain insight into the high-throughput microfluidics field. However, the review paper seems to be poorly structured. There is no strong connection between the three major areas reviewed. The coverage of knowledge in each field is also not satisfactory. Reflection and discussion of the reviewed work seem superficial. Overall, I think this paper is not qualified for publication in Micromachines.
1)The English should be significantly improved. The authors should pay attention to grammar and vocabulary to better convey their message. The language issue is preventing the fluent flow of logic.
2)In terms of chip materials, the authors only reviewed polymer-based platforms. However, I consider 3D printing compatible materials and glass-based microfluidic chips also hot materials under investigation.
3)There is quite an amount of reviews regarding the mixing and separating techniques for microfluidic chips in the literature. For example, [Thoriq Salafi, et al., Lab Chip, 2017,17, 11-33], [Lee CY, et al., Int J Mol Sci. 2011;12(5):3263-87] and [Nguyen, N.-T., et al., /Micromachines/*2017*, /8/, 186.]. Furthermore, the existing literature seems to have a broader coverage of technologies for microfluidics mixing and separating since the active and passive mixing/separating were distinguished and discussed. How the authors justify their intellectual input in reviewing the mixing and separating technologies in this paper?
4)With regards to 3D cell culture and ECM mimic in microfluidics, I found other works, for example, [Anal. Methods, 2019,11, 4220-4232] and [Li XJ, Valadez AV, Zuo P, Nie Z. Bioanalysis. 2012;4(12):1509–1525], more educating where not only the materials but also matrix formation methods and potential applications are thoroughly discussed.
Author Response
Dear Reviewer,
Thanks very much for your comments!
The manuscript has been modified greatly according to the comments. Firstly, for being more accurate to consistent with the content, the title was adjusted to “The fabrication and application mechanism of microfluidic systems for high throughput biomedical screening: A review”, because all the cases used in manuscript is about biomedical scope, and the mechanism is focus on application mechanism. The original three aspects are transform into two aspects: (1) The fabrication of high throughput biomedical microfluidic systems (HTBMS)(line number 53), including the scaffold materials and manufacturing methods for HTBMS (line number 60) and the construction of 3D microenvironment in HTBMS (line number 126); (2) The application and mechanism of HTBMS (line number 216), including biomedical synthesis and separation in HTMS without matrix environment (line number 224) and the application of droplet and cell/organ based drug screening of HTBMS with matrix environment (line number 392). A new figure (Figure 5) was added in the reversed manuscript to highlight the biomedical application of HTBMS with ECM (line number 400), and the table (Table 1) is modified into “The display and compare the throughput and efficiency of diverse mechanism for capturing CTCs from mimicking/real blood samples” (line number 390). The specific content has also been modified according to comments and suggestions. Please see the attachment for the detail response, thanks!
Regards,
Zhigang Hu

Reviewer 2 Report
This review article titled “The fabrication and mechanism of microfluidic systems for high throughput screening: A review” does not have a point of focus. Microfluidic systems do not always offer high-throughput performance in applications. The discussed aspects in sections 2 and 4, i.e. the materials for chip fabrication and for stimulating ECM thus have not so much related to high throughput. In section 3, the two main techniques for mixing and separation are potentially useful but they were not originally invented for high-throughput analyses. Low mixing efficiency is an intrinsic problem in microchannels which is contradictory to high-throughput analyses. Cell separation is a unique technique that microfluidics offers but they are mainly used for capturing rare cells from liquid biopsies. The authors have just listed a bunch of examples in each category of these two techniques without thoroughly reviewing and comparing the throughputs in terms of sample introduction rate, mixing rate, and analysis rate, etc., especially to the conventional non-microfluidic methods. Lacking such commentaries in this review article fails to reveal the connection between microfluidics and high throughput screening.
Author Response

(The authors gave the same response as above.)

Reviewer 3 Report
In this manuscript, Hu et al., provide an overview of microfluidics-based high-throughput screening systems. I think the paper is clearly presented and some of the very recent research results are included. However, I think there is still room for improvement. In particular, the title of the paper implies that the authors should focus more on high-throughput screening. Here, I am writing some of my comments as follows:
- In chapter two, in addition to the materials for the construction of the microfluidic chip alone, I suggest talking about the manufacturing methods altogether, since some of the materials rely on specific fabrication techniques for them to be molded into a device with a designed structure. Besides, what kind of requirement is needed particularly for high-throughput screening applications should be claimed.
- Chapter three is not sufficient for all kinds of microfluidic manipulations in a high-throughput screening system. For this, I recommend the authors cover “droplet microfluidics”, as an important subject in the high-throughput analysis. Some of the references can be tracked as follows:
Emerging droplet microfluidics, Chemical Reviews 2017, 117, 7964-8040;
Hydrogel Droplet Microfluidics for High-Throughput Single Molecule/Cell Analysis, Accounts of Chemical Research 2017, 50, 1, 22-31;
Droplet-based microfluidics in drug discovery, transcriptomics and high-throughput molecular genetics[J]. Lab on a Chip, 2016, 16, 1314-1331.
Besides, the figures for illustrating microfluidic microfiltration, Dielectrophoretic, Magnetophoresis, and Acoustophoresis are suggested.
- Chapter four: Instead of focusing on the matrix materials for fabricating ECM-mimetic environment, I would like to suggest the authors paying more attention to the construction of the entire system such as microfluidics-based 3D cell culture system, organ-on-a-chip systems, and drug screening platform, how to recapitulate the physicochemical microenvironments at multi-cellular and tissue-scale, to list a few.
Overall, I think the topic is good. However, I think both the structure and the content need to be improved.
Author Response
Dear Reviewer,
Thanks very much for the yourr comments and suggestion!
The manuscript has been modified greatly according to the comments. Firstly, for being more accurate to consistent with the content, the title was adjusted to “The fabrication and application mechanism of microfluidic systems for high throughput biomedical screening: A review”, because all the cases used in manuscript is about biomedical scope, and the mechanism is focus on application mechanism. The original three aspects are transform into two aspects: (1) The fabrication of high throughput biomedical microfluidic systems (HTBMS)(line number 53), including the scaffold materials and manufacturing methods for HTBMS (line number 60) and the construction of 3D microenvironment in HTBMS (line number 126); (2) The application and mechanism of HTBMS (line number 216), including biomedical synthesis and separation in HTMS without matrix environment (line number 224) and the application of droplet and cell/organ based drug screening of HTBMS with matrix environment (line number 392). A new figure (Figure 5) was added in the reversed manuscript to highlight the biomedical application of HTBMS with ECM (line number 400), and the table (Table 1) is modified into “The display and compare the throughput and efficiency of diverse mechanism for capturing CTCs from mimicking/real blood samples” (line number 390). The specific content has also been modified according to comments and suggestions. Please see the attachment for the detail response, thanks!
Regard,
Zhigang Hu

Round 2
Reviewer 1 Report
The manuscript has been significantly improved in terms of structure. However, there is still need for extensive editing for language and style before publication.
Reviewer 2 Report
The authors have significantly improved the manuscript. Since this review is oriented to be focused on biomedical screening, high-throughput diagnostic methods should be included.